# The Effects of Preterm Birth on Musculoskeletal Health-Related Disorders

**DOI:** 10.3390/jcm10215082

**Published:** 2021-10-29

**Authors:** Florian Schachinger, Sebastian Farr

**Affiliations:** Department of Pediatric Orthopaedics and Foot and Ankle Surgery, Orthopaedic Hospital Speising, Speisingerstrasse 109, A-1130 Vienna, Austria; schachinger.florian@gmail.com

**Keywords:** preterm birth, metabolic bone disease, fracture, developmental dysplasia of the hip, cerebral palsy

## Abstract

Preterm birth is associated with various diseases and conditions which demand multidisciplinary medical care. Approximately 10% of all neonates are born prematurely with an increasing survival rate in almost all Western countries. This ongoing, yet desirable trend is creating new challenges for sufficient medical treatment regimens, which should be upheld throughout the patients’ lives. Orthopedic surgeons are focused on musculoskeletal disorders and the improvement of patients’ ability to cope with the challenges of everyday life. The most common conditions associated with preterm birth are cerebral palsy and a dysregulation of the calcium/phosphorus metabolism, which may lead to fractures. These diseases may vary greatly in their organic manifestation and clinical presentation. This demands multidisciplinary cooperation and parental support. Clinical management is aimed on the early enhancement of a patient’s physical, as well as neurological condition, and to prevent the development of secondary musculoskeletal disorders. In this article, we give an overview of the current literature on the most common musculoskeletal disorders associated with preterm birth and critically discuss state of the art diagnostic standards and treatment algorithms.

## 1. Introduction

Preterm birth (PTB) is defined by the time of delivery before the 37th gestational week. Approximately 75% of all PTBs occur between the 34th and 36th gestational week [1]. Various risk factors have been identified. Intrauterine infection, history of previous preterm births, short cervical length and abnormalities of the growing fetus or the placenta are among the most common conditions for increasing the risk of spontaneous PTB [2,3,4,5,6]. In general, PTB is associated with diseases affecting the bronchopulmonary, cardiovascular and central nervous system [7] and a higher overall mortality risk [8]. Furthermore, preterm-born children may suffer from early onset musculoskeletal conditions which will be discussed in this review article. Since pediatric orthopedic surgeons focus on the treatment of conditions which impair the function of the musculoskeletal system, PTB-related conditions may have a significant impact on the further development and well-being of these individuals. The following sections will discuss the current evidence on preterm-related orthopedic disorders.

## 2. Preterm-Related Orthopedic Disorders

### 2.1. Cerebral Palsy

Cerebral palsy (CP) is the most common musculoskeletal disability associated with PTB. It was defined as “a group of permanent disorders of the development of movement and posture, causing activity limitation, that are attributed to nonprogressive disturbances that occurred in the developing fetal or infant brain” by Rosenbaum et al. [9]. The prevalence is approximately 1–2/1000 of all live births [10] with a significantly higher number in patients born preterm [11].

The cause often remains unknown, but may mainly be related to white matter brain damage. Little was the first author to describe the coherences of PTB and asphyxia, which were thought to be the underlying cause of CP [12]. Later, it was shown that, apart from low gestational age and low birth weight [13,14], perinatal infection is the most important risk factor. The presence of proinflammatory cytokines in intrauterine infections not only promotes PTB, but also the development of neuronal damage which amplifies the risk of developing CP [15,16]. Apart from intrauterine infection, every form of postnatal inflammation may be a contributing factor [17]. This is due to mitochondrial energy depletion amplified by oxidative stress which leads to necrosis or apoptosis of neurons and glia cells [18,19].

Nowadays, not only may the reliable diagnosis of birth asphyxia be difficult to make, but also the direct correlation as the leading cause of CP may be falsely acclaimed. According to a study conducted by Ellenberg and Nelson, less than 10% of all CP cases may be directly associated with birth asphyxia [20]. However, with a steady increase in gain of knowledge about the factors causing CP, a generally applicable prevention strategy has yet to be developed [21]. There is evidence on the neuroprotective effects of magnesium sulfate administration in women at risk for PTB prior to the 34th gestational week [22]. An overview review conducted by the Cochrane Library identified and summarized 15 systematic reviews which resulted in magnesium sulfate being the sole intervention with a high quality evidence of effectiveness which reduced the risk of CP in children [23]. Although, the mechanism of action is still not fully understood, animal studies showed a lower rate of apoptosis of neurons after ischemic brain injury [24] and decreased production of proinflammatory cytokines [25,26]. Furthermore, hypothermia showed promising results in late preterm infants within the first 6 h after birth by modifying the mechanisms of programmed apoptosis in neuronal cells and lowering neuronal metabolic rate to prevent energy depletion [27].

The symmetric muscle spasticity of the limbs, especially in the adductor and flexor muscles of the lower extremity is characteristic for CP patients. Most of them have problems with proper motor function. This may be caused by several factors. CP patients show a decreased ability in generating rapid muscle force and overall muscle strength [28,29] and a tendency for simultaneously contracting agonistic and antagonistic muscle groups [30]. This may cause dyskinesia, ataxia and athetosis in varying degrees. Furthermore, patients are prone to developing contractures and muscle stiffness. As previously believed, this is not caused by an overactivation of the affected muscle groups, but by an increased collagen type I presence in the surrounding connective tissue of the muscle fibers combined with a shorter sarcomere length [31]. Frequently, decreased proprioceptive abilities are present which add to the underlying problem.

Depending on the severity of disease, hearing and/or visual impairment, mental impairment, delay in psycho-cognitive development and even dysregulation of the cardiovascular and respiratory system may be present [32]. Even in rather mild cases there may be a significant impairment of fine motor skills which may not be apparent at first sight [33]. The diagnosis is made using clinical as well as radiological examination techniques with emphasis on examiner observations and parent reports [34,35].

The Gross Motor Function Classification System (GMFCS) works as a tool to assess the severity of CP [36,37] and to predict the further development in a specific patient [38]. However, the current literature shows that this is somewhat of a difficult endeavor. A significant amount of patients who were initially diagnosed with CP turned out to be false positive [39]. Many symptoms may become apparent or even more severe later on in life as the infantile brain is still in development. Even examinations performed by a healthcare professional yielded a certain false positive rate [40]. However, it has to be emphasized that CP is a nonprogressive condition.

Currently, there is an effort to use standardized parent-reported functional scores and gait patterns to determine whether the patient has CP. One independent parameter for CP is the step width at a corrected age of 20 months [41]. Modern MRI technology is one of the most important pillars for making the correct diagnosis and to detect white matter brain damage at an early stage [42,43] as it may be very difficult to find in early cerebral ultrasound examinations [44]. White matter brain damage is mostly caused by periventricular leukomalacia which in turn is caused by the inability of the infantile brain to increase the blood supply to hypoperfused border zones [45]. The neurological outcome correlates with the extent of white matter brain damage and therefore the severity of disease in CP patients [46]. Future motor function impairment may be reliably predicted by using diffusion tensor imaging and gait impairment patterns in preterm infants [47,48,49].

Treatment options in CP patients are various. In general, the goal is to improve quality of life and to ease daily caregiving by doing as little harm as possible (Figure 1). This may be achieved by a conservative treatment regimen and/or surgical intervention. The surgical concept rests on several pillars. First and foremost, the decision to opt for surgery should be made after adequate consultation of the treating physiotherapists, caregivers and parents. The goal is to improve the patients’ ability to perform activities of daily living and prevent further complications like joint contractures or dislocations. Spasticity may effectively be treated with botulinum toxin type A with a satisfactory mid-term outcome [50,51]. In cases with dystonia, intrathecal baclofen application was shown to be beneficial [52]. Depending on the underlying problem, surgical procedures normally consist of derotational osteotomies [53,54], open reduction of hip dislocations with or without concomitant bony procedures [55,56,57], tenotomies [58,59,60,61,62] or guided growth with temporal (hemi) epiphysiodesis [63,64,65,66]. Care has to be taken with tenotomies for gaining more range of motion (ROM) of the affected joints which is generally accompanied by a further weakening of the muscle in our clinical experience. As every form of CP is characterized by muscle weakness rather than spasticity, this has to be carefully taken into account [67].

### 2.2. Fractures

The majority of bone mineralization happens in the last trimester. Thus, an increased risk for pediatric fractures in preterm infants may be assumed [68]. Previously, it was thought that preterm infants were not at risk for sustaining fractures in the first months of age [69]. However, assault-related fractures were not included before a recent study by Michaud et al., which identified child maltreatment as a significant factor. In this cohort study, PTB was identified as an isolated risk factor for hospitalization due to pediatric fractures. The examined population showed a significant number of assault-related fractures, which mainly happened before 18 months of age. The highest incidence of fractures which required hospitalization was shown in infants born between the 32nd an 36th gestational week [70]. Contributing factors may be family related and due to low socioeconomic status [71,72]. Furthermore, preterm infants who are considered small at gestational age may have a lower bone mass density in adulthood than those who had an appropriate size, despite being born preterm [73]. This may further increase the risk of certain facture types in the further course of life.

### 2.3. Metabolic Bone Disease

Metabolic bone disease (MBD) is a hypernym for various diseases and syndromes with impaired calcium and phosphorus balance. Patients born before the 27th gestational week are mainly affected as most of the bone mineralization happens after this point [68]. Symptoms vary from decreased trabecular bone density to fractures after minimal trauma, slower growth rate and decreased height. Early clinical diagnosis is difficult whereas elevated parathormone and alkaline phosphatase blood levels and decreased calcium and phosphorus serum levels may act as early indicators. With an approximate incidence of 7% in preterm infants, it may be overlooked rather easily [74]. A recent study isolated three independent risk factors for the development of MBD in preterm infants: a gestational age under 30 weeks, vitamin D supplementation delay over 14 days of age and the start of pure enteral nutrition after 28 days of age [74]. Furthermore, an early transition to total enteral nutrition should be sought by the treating physicians.

### 2.4. Developmental Dysplasia of the Hip

Developmental dysplasia of the hip (DDH) is a hypernym for an abnormal acetabular shape of the neonatal hip which may lead to increased wear and early osteoarthritis and severe functional gait impairment in the case of undetected dislocation. 

The overall incidence of developmental dysplasia of the hip (DDH) shows a massive regional variance. In Scandinavia and Western Europe there are reports ranging from 0.66–39.4 cases/1000 births [75]. The overall incidence in Austria is approximately 0.24% of all live births [76]. Well-known risk factors for developing DDH are positive family history, breech position, primiparity [77], oligohydramnios [78] and genetic predisposition, however evidence of the latter is low [79].

Symptoms that patients may present with are positive Barlow and Ortolani sign, abduction deficit, torticollis, club foot deformity and female gender [80,81,82]. The current diagnostic standard is the hip sonographic screening method which was first described by Graf in the early 1980s [83,84]. Graf graded the severity of the developmental retardation of the hip according to the shape of the joint and its corresponding alpha angle.

In the early stages, an ultrasound screening was recommended to be performed within the first 2 weeks after birth [85,86]. In current clinical practice, an ultrasonographic examination within the first week after birth and a follow-up examination at 6 weeks is recommended [87,88].

It was long assumed that PTB was associated with DDH [89,90,91]. However, even early reports show that it may be prognostically beneficial against developing DDH [92]. This may be due to the primary presence of mechanical risk factors during the last trimester of pregnancy. In this period, the intrauterine space decreases due to the growth of the fetus which may result in adduction of the femur on the side of the maternal spine [93,94]. A recent biomechanical study shows that physical stimulation of the fetus’ hip joint (moving, kicking) is essential for intrauterine joint development. The presented data shows a decreased range of motion and overall strength of the fetus’ kicks in breech position and cases with oligohydramnios which may cause a retardation in hip development [95]. 

A physiological hip joint is consistently present within the 31st gestational week with decreasing alpha angles thereafter [96]. Further studies confirm this theorem with preterm infants showing better alpha angles than term infants and a lower presence of immature hips [94,97]. In a large Scandinavian cohort study, almost all patients had either a fully developed hip joint or a slight maturation deficit (Graf Type IIa) [94]. However, even with PTB not affecting DDH, preterm infants may be prone to developing osteoarthritis of the hip joint with a subsequent need for total hip arthroplasty [98].

## 3. Conclusions

The survival of preterm infants has improved dramatically over the last decades. Thus, a steadily rising number of patients who are in demand for specific treatment regimens tailored to their needs has to be expected in the future. This has to be considered not only by orthopedic surgeons, but neurologists, pediatricians, physiotherapists, occupational therapists, orthopedic technicians and many other specialties alike. First and foremost, the goal should be to prevent the development of CP. There is strong evidence on the neuroprotective effects of magnesium sulfate application in women at risk of preterm delivery and on the efficacy of postnatal cooling in infants with hypoxic encephalopathy. Modern MRI technology has enabled us to further understand the specific pathologic pathways in the development of CP and has allowed us to attain a prognosis of future neurological impairments. With this in mind, a treatment strategy should be developed at the earliest possible time after the diagnosis has been made [99]. Furthermore, psychological support of preterm infants’ families should be sought at an early point after gestation. Although parents should not be under general suspicion, emergency room physicians must always take the possibility of child abuse into consideration as preterm infants are at higher risk for sustaining abuse-related injuries.

## Figures and Tables

**Figure 1 jcm-10-05082-f001:**
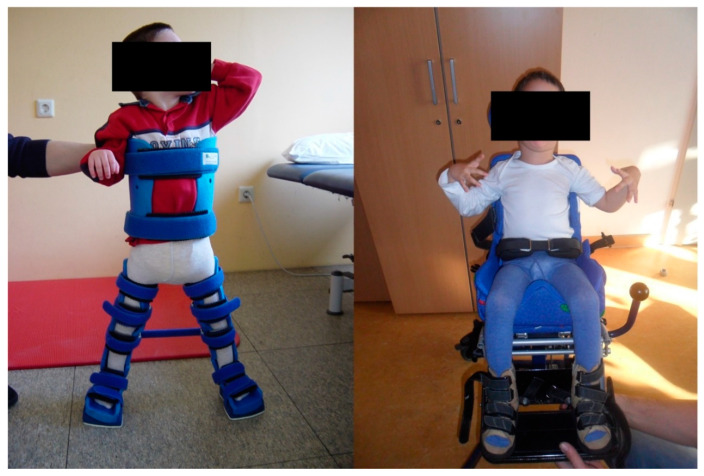
CP patients with a supportive orthosis for the trunk and lower limbs (**left image**) and in a wheel chair with typical upper limb posture (**right image**) are shown.

## Data Availability

Not applicable.

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
