# Peer review of "The Effects of Preterm Birth on Musculoskeletal Health-Related Disorders"

_jcm, 2021, doi:10.3390/jcm10215082_

Round 1

Reviewer 1 Report

The aim of the paper is to give overview of the current literature on the most common musculoskeletal disorders associated with preterm birth and critically discuss state of the art diagnostic standards and treatment algorithms. 

However, almost no surgical procedures are discussed in details throughtout the manuscript. Moreover, authors are pediatric ortopaedics and would be very exciting to know their experiences, number of pretem infants indicating for surgery and their surgical outcome. 
I recommend to complete surgical procedures mainly dealing with Cerebral Palsy complication (not only the list of potentially suitable surgeries as noted in the manuscript).

Author Response

First, we would like to give our thanks to the reviewer for the efforts of reviewing our manuscript and the constructive comments and suggestions.

Replies to Reviewer 1:

No surgical procedures are discussed in detail.

Response: Thank you for your comment. As this review was meant to be an overview article of current pediatric orthopedic treatment options for preterm associated conditions in a special volume for obstreticians and gynaecologists we refrained from going into too much detail for each surgical procedure. We acknowledge a certain need for a compendium on such a topic, but chose to keep it short and simple for the ease of reading as these procedures are normally performed by highly specialized and experienced surgeons.

Reviewer 2 Report

This review summarizes current knowledge around musculoskeletal lesions/disorders related to Preterm birth. The review is a little short and section must be developed to better understand the underlying mechanisms.

Please find hereafter some suggestions for the authors:

  • Introduction is quite short and a global presentation of risk factor for preterm birth and known mechanisms are not presented. However, Authors highlighted from line 64 to 74, inflammation as a cause to preterm birth. This should be move to introduction.
  • There is a lack of references to allow readers to have more info about underlying cellular mechanisms. Ref #13 is a little old and some authors have published more recent reviews about this research field.
  • There is a lack of references and explanation about how muscles are affected by preterm birth. Line 85 should be extended.
  • Line 108: authors do not explain why study WM is important to PTB. Maybe try to highlight this when respond to precedent comments
  • PTI and PTB are confusing abbreviation. “Preterm Infant” can be written to easily read this interesting review
  • line 131: lack of references

Author Response

First, we would like to give our thanks to the reviewer for the efforts of reviewing our manuscript and the constructive comments and suggestions.

Replies to Reviewer 2:

A global presentation of risk factors for preterm birth and mechanisms are not presented.

Response: Thank you very much for reviewing our manuscript and providing us with your valuable input. We added a collection of the most common and well known risk factors in the introduction.

However authors highlighted from line 64 to 74, inflammation as a cause of preterm birth. This should be moved to introduction.

Response: Thank you for your remark. We agree that intrauterine infection is one of the most important factors for causing preterm birth, whether it’s caused by a preterm premature rupture of the membranes or idiopathic, but the goal of our statement was not to highlight it as an already well known risk factor. The studies by Dammann et al. and Leviton et al. depict the theorem of pro-inflammatory cytokines promoting brain damage. Therefore, we’d prefer to keep these very important findings in context. However, we added a section dedicated to the most common risk factors for preterm birth in the introduction as mentioned above.

There is a lack of references to allow readers to have more info about underlying cellular mechanisms.

Response: Thank you for your advice. This review article was written for an OB/GYN special volume and aimed to focus onto the pediatric orthopedic treatment of included but not limited to cerebral palsy. The target group consists of physicians, physiotherapists and orthopedic technicians who work closely with affected patients in a clinical setting. Therefore, we refrained from going into too much detail on molecular and biochemical processes.

Ref #13 is a little old and some authors have published more recent reviews about this research field.

Response: We added a recently published review article in the reference list.

There is a lack of references and explanation about how muscles are affected by preterm birth.

Response: Thank you for your suggestion. We extended the paragraph covering pathologic gait patterns and added essential information on the cellular changes of the muscles of CP patients.

Line 85 should be extended

Reponse: We extended the paragraph and added more detailed information.

Line 108: authors do not explain why study WM is important to PTB. Maybe try to highlight this when respond to precedent comments.

Response: Thank you for your advice. We further elaborated the physiological and neuroanatomic background and the importance of white matter damage in CP patients.

PTI and PTB are confusing abbreviations. “Preterm infant” can be written to easily read this interesting review.

Response: We changed all of the abbreviations containing “PTI”.

Line 131: Lack of references.

Response: There was no intention of adding a specific reference to this statements as this observation is based on our practical experience. We rephrased our statement for further clarification.

Round 2

Reviewer 1 Report

The authors made substantial progress 

Author Response

Many thanks for your help to improve this paper. S Farr